# LDA filter: A Latent Dirichlet Allocation preprocess method for Weka

P. Celard[1,2,3]*, A. Seara Vieira[1,2,3], E. L. Iglesias[1,2,3], L. Borrajo[1,2,3]

**1** Computer Science Dept., Univ. of Vigo, Escuela Superior de Ingeniería Informática, Ourense, Spain, **2** CINBIO - Biomedical Research Centre, Univ. of Vigo, Vigo, Spain, **3** SING Research Group, Galicia Sur Health Research Institute (IIS Galicia Sur), SERGAS-UVIGO, Vigo, Spain

* pedro.celard.perez@uvigo.es

**Data Availability Statement:** Relevant experiments results data is available within the manuscript and all data sets used are available from the Figshare repository (URL: https://figshare.com/articles/dataset/LDA_Weka_Datasets_zip/13047917).

## Abstract

This work presents an alternative method to represent documents based on LDA (Latent Dirichlet Allocation) and how it affects to classification algorithms, in comparison to common text representation. LDA assumes that each document deals with a set of predefined topics, which are distributions over an entire vocabulary. Our main objective is to use the probability of a document belonging to each topic to implement a new text representation model. This proposed technique is deployed as an extension of the Weka software as a new filter. To demonstrate its performance, the created filter is tested with different classifiers such as a Support Vector Machine (SVM), k-Nearest Neighbors (k-NN), and Naive Bayes in different documental corpora (OHSUMED, Reuters-21578, 20Newsgroup, Yahoo! Answers, YELP Polarity, and TREC Genomics 2015). Then, it is compared with the Bag of Words (BoW) representation technique. Results suggest that the application of our proposed filter achieves similar accuracy as BoW but greatly improves classification processing times.

## Introduction

Digitization of documents has increased exponentially in recent years. Books, articles, websites or publications made by users on different platforms are some examples of text content that is stored minute by minute. As the number of documents increases, the task of locating and accessing this content becomes more difficult. It is necessary to use tools for automatic classification to address this problem.

Weka is a workbench that provides an environment for automatic classification and data mining. It includes a set of methods for data mining tasks such as classification, clustering, attribute selection, regression and association [1]. It has been used in multiple researches and facilitates the usage of data processing tools to the user [2]. The Weka environment also includes different tools for data transformation and preprocessing. However, in the field of text classification, Bag of Words (BoW) is the only document representation method that can be applied in Weka. The bag of words approach represents every document by a vector where elements describe the weight or relevance of the words in the document. Usually, the weight

**Funding:** We also appreciate the support provided by Consellería de Educación, Universidades e Formación Profesional (Xunta de Galicia) under the scope of the strategic funding of ED431C2018/55-GRC Competitive Reference Group. The funders had no role in study design, data collection and analysis, decision to publish, or preparation of the manuscript.

**Competing interests:** The authors have declared that no competing interests exist.

represents the frequency (number of occurrences) of that word in the document [3]. This is the most common approach to represent documents in text classification [4].

BoW relies on the number of appearances of words in the documents. This idea is seen on traditional search engines. These systems look for documents that contain one or more occurrences of the words that the user specifies. This method can obviate valid results that do not contain the required word even if the topic is related to the original query. This is a problem, since in most cases, the input query is based on a topic, not on a specific word.

The technological advances of recent years and new research on machine learning provide tools to process a high number of documents using algorithms that extract information from their original texts. One of these algorithms is Latent Dirichlet Allocation, better known by its acronym LDA [5], which offers the possibility of discovering a set of topics in a group of documents. Multiple authors have successfully put it in practice.

Blei *et al.* [6] implement document modelling, text classification and collaborative filtering techniques using LDA. They conclude that LDA slightly reduces classification performance but improves overall efficiency thanks to its dimensionality reduction characteristic.

Wang *et al.* [7] merge the representation of documents using LDA applying labels to improve the performance of text classifiers, modifying it to work as a semi-supervised LDA method which is able to incorporate partial expert knowledge at both word level and document level. Their results conclude that accuracy is increased as more documents are labeled, making it a viable option to be of use in real-world applications where training corpora may be large, but only a small portion of expert knowledge (labels) is available for training.

Kim *et al.* [8] use LDA combined with TF-IDF and Doc2Vec to implement a Multi-Co-Training (MCT) system, increasing the variety of feature sets for document classification. The experimental results demonstrate that the proposed MCT is robust to parameter changes and outperforms benchmark methods under various conditions.

Jin et al. [9] propose a model called DLDA assuming that the knowledge acquired through long texts can help to extract topics from short texts. To achieve this they use two sets of LDA topics, one called "target" based on short texts and another called "auxiliary" created from long texts. Their experimental results show that their dual model outperforms basic LDA and helps to improve short text clustering.

Zhou et al. [10] propose a text representation model that combines the word embedding technique Word2Vec and LDA. Based on their experiments they conclude that accuracy is improved and that LDA offers a solution to problems of high dimensionality and high sparsity caused by BoW models.

Quan et al. [11] use a Self-aggregation based topic model (SATM) that merges clustering and LDA. The experimental results conclude that the proposed SATM model can distill more meaningful topics and improves accuracy when applied to short texts.

Cheng et al. [12] use LDA and word co-occurrence patterns (i.e., biterms) in the corpus to detect topics. Each biterm is addressed as a semantic unit that exhibits a single topic, detecting the words most likely to be together. Their model improves the coherence of topics and this is reflected in the system performance.

Pröllochs and Feuerriegel [13] propose an automatic text mining framework to analyse texts as financial disclosures from firms. They use LDA as a topic modelling technique to reveal the strengths and weaknesses of a firm identifying business units, activities and processes subject to risks. Their proposed framework automatically engages in granular recommendations at different business levels, highly improving existing business management tools (e.g., SWOT analysis).

Despite being a widely used framework, currently Weka does not include packages that allow to represent texts using LDA. Based on the previous studies, the main objective of this

work is to create an LDA text representation model and implement it into the Weka suite, providing also an analysis about its performance and feasibility. The usage of LDA topics as an alternative to classic representation models, such as the bag of words, allows to classify documents in a more efficient way since it greatly reduces the number of terms that classifiers should work with.

## Latent Dirichlet Allocation

Latent Dirichlet Allocation is a statistical model that implements the fundamentals of topic searching in a set of documents [5]. This algorithm does not work with the meaning of each of the words, but assumes that when creating a document, intentionally or not, the author associates a set of latent topics to the text. For example, the document shown in Fig 1 deals with the structure and function of the heart. Throughout the text, words such as *heart*, *basal* and *arteries* that relate to the topic *heart structure* stand out; words like *contraction* and *shell* are related to the topic of *muscle functioning*; and finally, the words *urine* and *blood* are related to the topic *functions of the liver*. If the entire document is analyzed in the same way, it could be concluded that it deals with the parts of the heart, muscle functioning and liver functions in different proportions.

As seen in Fig 1, LDA is a statistical model that tries to capture this intuition. It considers that each document covers subjects in different proportions, and each word in a document corresponds to one of those subjects.

The LDA model assumes that each corpus of documents deals with a set of predefined topics. Topics are formally described by a distribution over the entire vocabulary, as shown

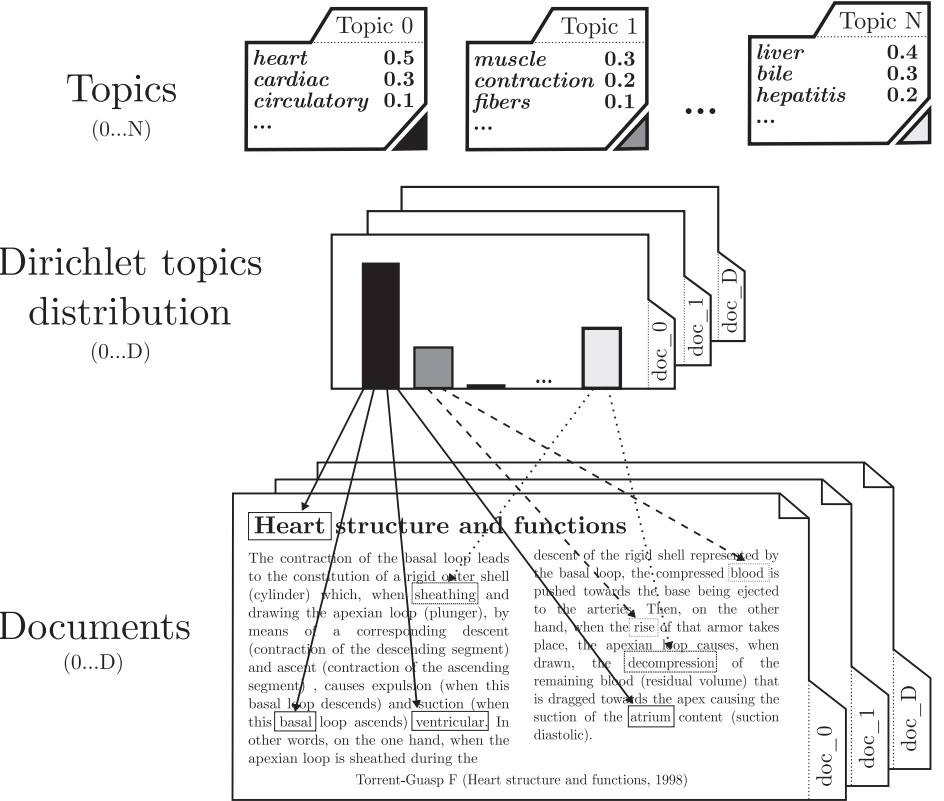

**Fig 1. LDA underlying intuition.** Generation of documents through topics following the Dirichlet distribution.

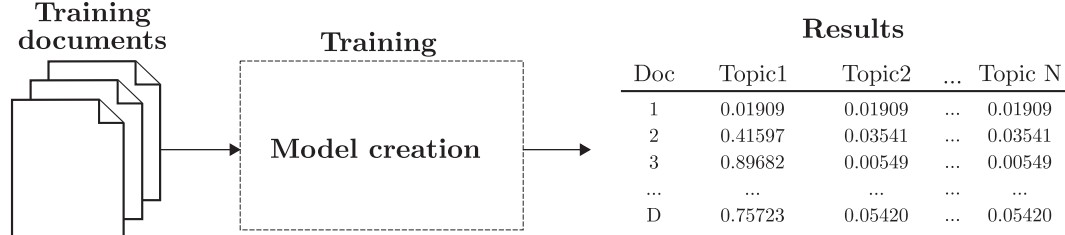

**Fig 2. LDA workflow.** LDA training and inference phases. *D* represents the total number of documents of the corpus and *N* the number of topics.

in Fig 1. LDA assumes that each document in the collection is created in two steps. Firstly, a distribution on the topics (histogram of the Fig 1) is chosen for that document. In the example, the topics of *heart structure*, *muscle function*, and *liver functions* are chosen. Secondly, for each place that could contain a word in the document, a random topic and a word of its corresponding distribution is assigned. The process is repeated for the entire corpus. Thus, the main characteristic of LDA arises: all documents share the same topics in different proportions.

Based on this theory, the LDA model can be applied to an input collection of documents in order to infer their latent topics by using the so-called Gibbs Sampling algorithm [5]. In this process, the algorithm iterates through the words of all the documents and calculates the most representative word for each topic. The words can appear multiple times in the same document and be repeated in different documents at the same time. The algorithm is able to modify the topic that best represents them in each iteration.

As shown in Fig 2, a model is built after applying the Gibbs Sampling with a training collection, generating a topic distribution for each document.

## Materials and methods

### Implementation

The main objective of this work is to create and to implement an LDA representation model as an extension of the free Weka software, providing a new filter for document corpora. In addition, we analyze the application of this document representation model to improve the efficiency of current text classification algorithms.

Two of main Weka elements are filters and classifiers. Filters collect data, transform it and generate a new set of features/attributes to represent it. After applying a filter, the resultant data can be used as input to train and test classification algorithms.

Our filter, named LDA_Filter is an unsupervised preprocessing method that uses the probability of a document belonging to each topic to represent that document. These probabilities are obtained through the LDA topic modeling algorithm which seeks to discover the internal structure of documents by inferring their topics.

LDA_Filter becomes a part of the Weka tool, bringing together different algorithms for preprocessing and text classification. Weka allows the user to apply this new filter to different sets (corpus) of documents, and test them easily with different types of classification algorithms already implemented in the tool. In addition, Weka greatly facilitates the parametrization of the filter, thus allowing to obtain the best representation for a corpus.

This new filter is implemented using an open source library called MALLET (Machine Learning for Language Toolkit) [14], a free use tool developed by Andrew McCallum in collaboration with multiple members and students from the University of Massachusetts Amherst

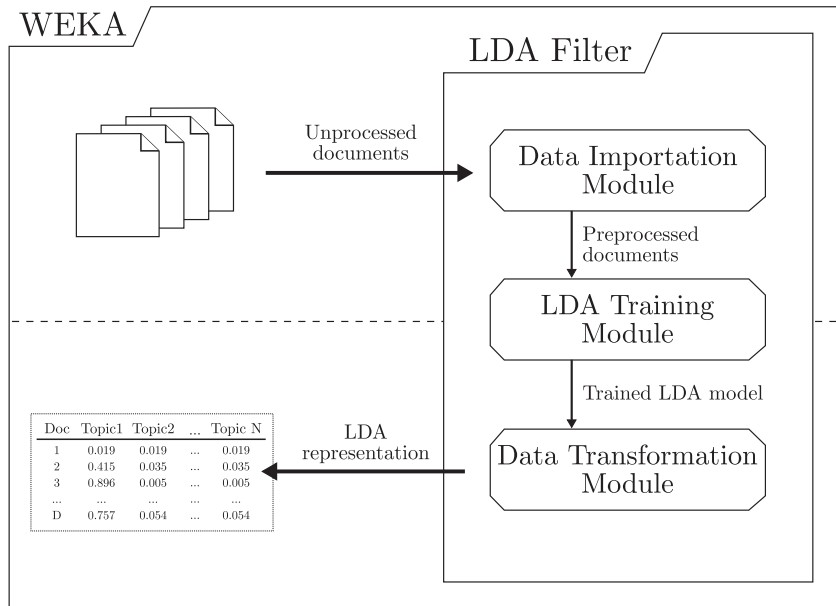

**Fig 3. Workflow and data transfer scheme.**

and the University of Pennsylvania. MALLET offers multiple techniques for statistical processing of natural language, document classification, clustering, topic modeling, information extraction and other machine learning applications to text documents.

As shown in Fig 3, LDA_Filter is implemented as a package containing three modules: one for data import, another for LDA model training, and the last for data transformation. The internal process is carried out without user interaction, so once the parameter values are indicated and the process is started, a new representation of the documents will be obtained.

LDA_Filter workflow consists of the following steps. Firstly, the training data is imported and an LDA model is trained in order to infer the topic and vocabulary distributions. Then, the trained model is used to represent the collection of documents with a set of attributes which dimensionality is equal to the number of topics specified by the user. Each document is then stored as an array where each element indicates the probability of that document belonging to a topic.

The proposed filter also preprocesses the data, applying stopwords and stemming if desired. Table 1 shows all the parameters that can be modified to vary the behaviour of the filter.

## Case studies

### Use of the tool

The filter has been adapted to take advantage of the versatility of the Weka package installer, allowing a simple installation and facilitating its usage as much as possible. Its usage, in conjunction with Weka tools, allows the study of its application on different sets of documents, classification techniques and general machine learning tools.

In order to use our filter in Weka, it can be accessed through the drop-down menu in the FILTERS → UNSUPERVISED → ATTRIBUTE → LDA section (see Fig 4). It can also be used through Java code by creating a new object of the LDA class.

**Table 1. LDA attributes.**

| Attribute | Description |
|---|---|
| alpha | Smoothing parameter for document-topic distributions. *alpha* = [*thisvalue*]/[*numtopics*] (default: 0.5) |
| beta | Smoothing parameter for each topic. (default: 0.01) |
| burnIn | The number of iterations to run before first estimating Dirichlet hyperparameters. (default: 200) |
| numIterations | The number of iterations of Gibbs sampling. (default: 1000) |
| numThreads | The number of threads for parallel training. (default: 1) |
| numTopWords | The number of most probable words to print for each topic after model estimation. (default: 20) |
| numTopics | Specifies number of topics (default: 10) |
| stemmer | The stemming algorithm (classname plus parameters) to use. (default: null) |

LDA adjustable parameters.

As shown in Fig 5, the user can edit the value of the parameters of our filter including number of topics, iterations, alpha, beta, etc. Modifying them changes the way our filter works and the final attributes that its execution returns. For greater detail on the parameters, see Table 1.

As explained in the previous section, the number of returned attributes changes adapting to the number of topics indicated. If the number of topics is set to 10, as can be seen in Fig 6, the entire document collection will be represented with 10 features/attributes.

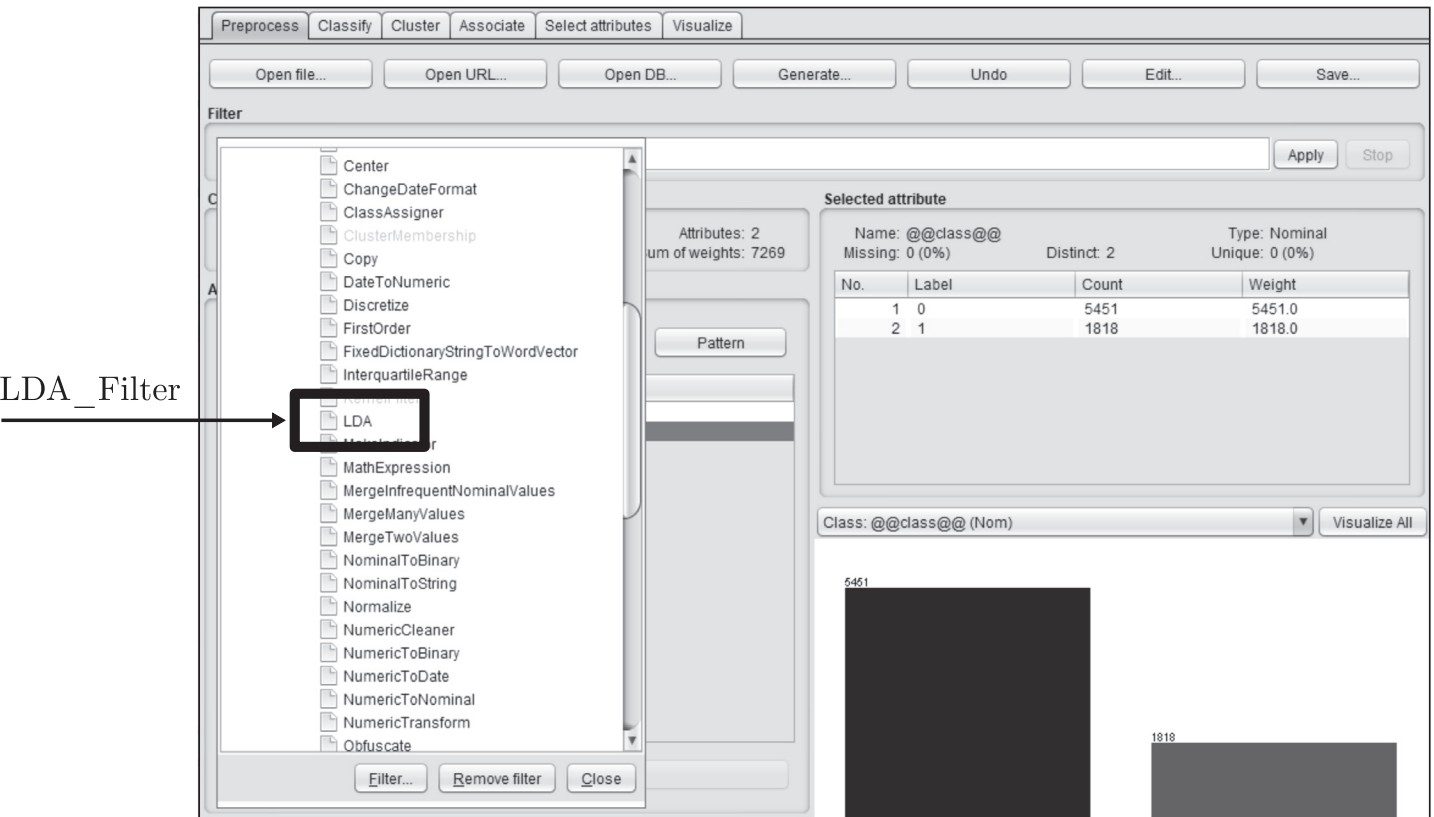

**Fig 4. Weka filter selection.** Select the filter from the filter drop-down list.

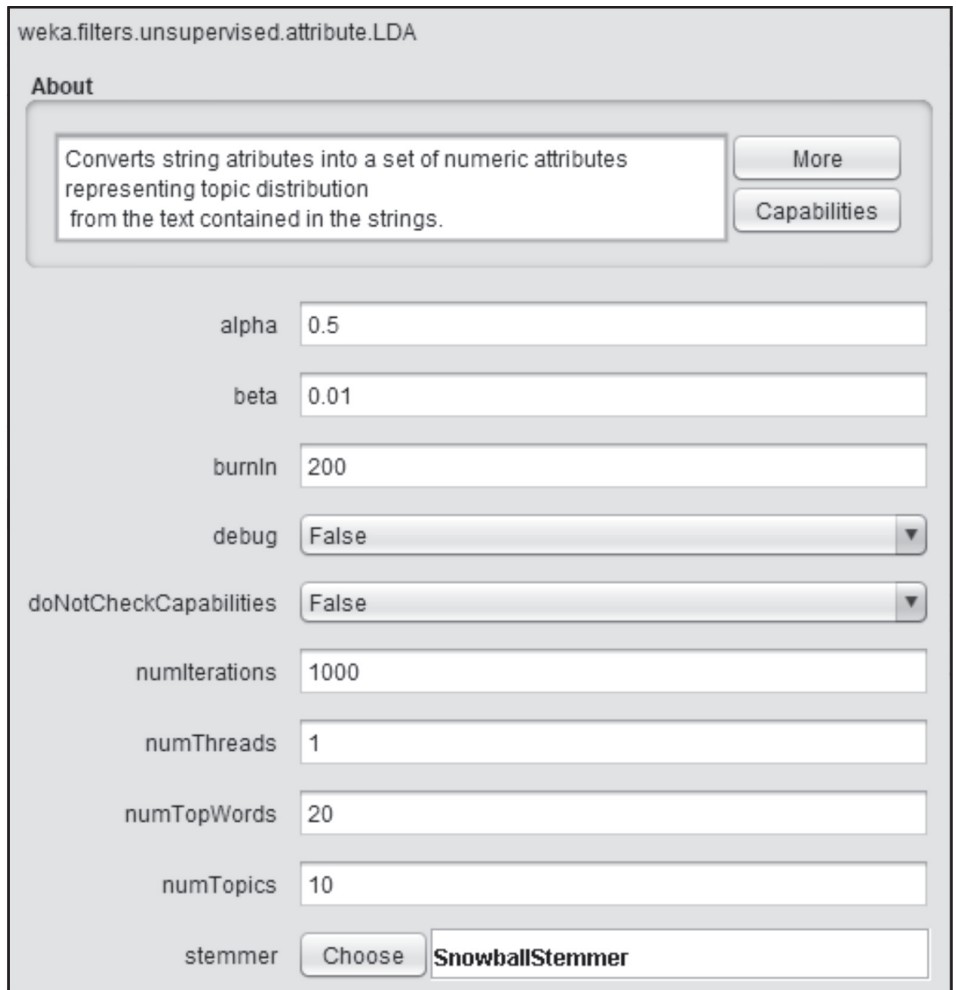

**Fig 5. Parameter value selection.** Modify the default value of the filter parameters.

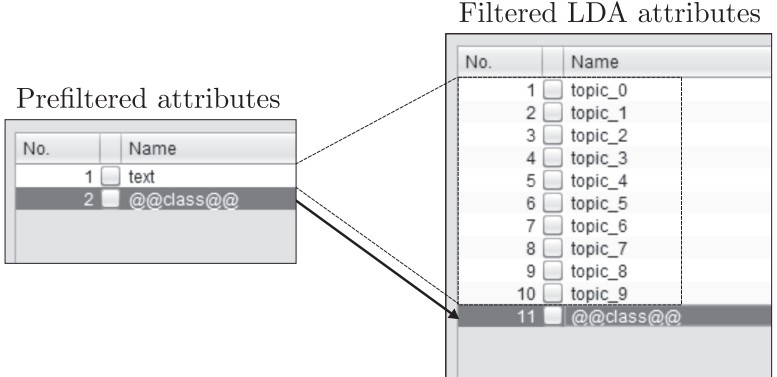

**Fig 6. Result of the filter application.** An attribute is generated for each topic.

**Table 2. Classifier result.**

| Summary | | | | |
|---|---|---|---|---|
| Correctly Classified Instances | | 2118 | 85.71% | |
| Incorrectly Classified Instances | | 353 | 14.28% | |
| Kappa statistic | | 0.57 | | |
| Mean absolute error | | 0.14 | | |
| Root mean squared error | | 0.378 | | |
| Relative absolute error | | 38.13% | | |
| Root relative squared error | | 87.55 | | |
| Total number of instances | | 2471 | | |

| Detailed Accuracy By Class | | | | | | |
|---|---|---|---|---|---|---|
| | TP Rate | FP Rate | Precision | Recall | F-Measure | Class |
| | 0.953 | 0.433 | 0.870 | 0.953 | 0.909 | 0 |
| | 0.567 | 0.047 | 0.798 | 0.567 | 0.663 | 1 |
| Weighted Avg. | 0.857 | 0.337 | 0.852 | 0.848 | 0.589 | |

Result obtained after applying an SVM classifier to the LDA representation.

As soon as the results of applying our filter are obtained, a classifier can be applied. Table 2 shows a brief example of the result of applying an SVM classifier in Weka.

## Experiments

**Datasets.** To demonstrate the efficiency of the proposed filter, we present a set of experiments which have been performed on six different datasets.

The first data collection is the Reuters-21578 document corpus, originally organized in 90 different news categories. The subset used for this work contains documents from the 8 top-sized categories as used in [15], ending up with a total of 8,055 documents.

The second data collection is the 20 Newsgroup dataset. This is a collection of approximately 20,000 newsgroup documents, partitioned (nearly) evenly across 20 different newsgroups. It was originally collected by Ken Lang [16], and has become a popular dataset for experiments in text application of machine learning techniques.

The third data set is the OHSUMED collection, compiled by Hersh *et al.* [17]. It is a subset of the MEDLINE database, a bibliographic database of important medical literature maintained by the National Library of Medicine. OSHUMED contains 348,566 references consisting of fields such as titles, abstracts, and MeSH descriptors from 279 medical journals published between 1987 and 1991. The collection includes 50,216 medical abstracts from the year 1991, which were selected as the initial document set. Each document in the set has one or more associated categories (from the 23 disease categories). In order to build a binary class corpus, we consider the documents that have been assigned with the Neoplasms (C04) category as the relevant documents. The other categories are considered as the common bag of non-relevant documents.

One of the tasks in TREC Genomics 2005 Track [18] was to automatically classify a full-text document collection with the train and test sets, each consisting of about 6,000 biomedical journal articles. Systems were required to classify full-text documents from a two-year span (2002-2003) of three journals, with the documents from 2002 comprising the train data, and the documents from 2003 making up the test data. The categorization task assessed how well systems can categorize documents in four separate categories: A (Alelle), E (Expression), G

(GO annotation), and T (Tumor). In this paper, the Allele annotation category is used to test the performance of the proposed model, being the forth dataset used in the evaluation. Documents can be classified as relevant or non-relevant.

The Yelp Polarity dataset, consists of reviews from Yelp. It is extracted from the Yelp Dataset Challenge 2015 data. The Yelp reviews polarity dataset was constructed by Xiang Zhang [19] from the above dataset. It was created by considering star rating 1 and 2 negative, and 3 and 4 positive. For each polarity, 280,000 training samples and 19,000 testing samples were taken randomly. In total there are 560,000 training samples and 38,000 testing samples. Negative polarity is class 1, and positive class 2. It has become a popular dataset for text classification and sentiment analysis [20, 21] and it is used as the fifth dataset in this paper.

The Yahoo! Answers dataset includes 4,483,032 questions and their corresponding answers obtained through the Yahoo! Research Alliance Webscope program. The Yahoo! Answers topic classification dataset was also constructed by Xiang Zhang [19, 22], using the 10 largest main categories. Each class contains 140,000 training samples and 6,000 testing samples. Therefore, the total number of training samples is 1,400,000 and 60,000 in the case of testing samples. It is used as the sixth text classification benchmark in this research.

**Evaluation.** A set of tests were carried out applying different classification algorithms to analyze the performance of the proposed LDA-filter compared to the classic representation of bag of words (BoW). The classifiers were applied to the corpus mentioned in the previous section, creating for them an LDA representation and a BoW representation. With the purpose of observing how this new representation affects the performance of the classifiers, we analyze the behaviour of SVM, k-NN and Naive Bayes, which have proven their effectiveness in many scenarios [15, 23, 24].

In order to carry out the experiments, we had to adapt the data sets due to the limitations of the testing equipment. It has been necessary to reduce the number of documents in the data sets while maintaining the proportion between the classes. Specifically, the Yelp Polarity dataset, composed of two classes, is left with 28,520 documents, while Yahoo! Answers remains with 7,998 documents.

Initially, standard text preprocessing techniques are applied to each document. A predefined set of stopwords (common English words) is removed from the text, and a stemming algorithm based on the Lovins stemmer [25] is applied to the remaining words. Once the preprocessing phase is finished, a ten fold cross validation is performed, applying a set of classifying algorithms for each representation technique. Among the results obtained, the Kappa value and total execution times are measured.

Kappa represents the value of Cohen's Kappa coefficients, a statistical measure that indicates the agreement between different classifiers. It takes into account the possibility of casual successes, and can take values between -1 and 1, indicating the negative values that there is no agreement, and the values between 0 and 1 the level of existing agreement. Between 0.01 and 0.20 is a slight agreement, between 0.21 and 0.40 fair agreement, 0.41 and 0.60 moderate agreement, 0.61 and 0.80 substantial agreement and between 0.81 and 1 perfect agreement. Given these values, the larger the agreement, the more reliable the results of the classifiers [26].

To demonstrate that the observed results are not a casual effect of execution, we use a statistical test that measures the actual confidence of a given set of tests. A two-tailed Student t-test is executed on the Kappa values obtained by executing the classification algorithms on the representation of bag of words and LDA of each of the corpus. This type of test has an $\alpha$ significance value of 0.05, which means that if the result is less than that value, a statistical difference between the samples cannot be detected. If the value is higher, it is necessary to observe the average of the samples to know which method is better.

**Table 3. Best parameter values.**

| Parameter | SVM | k-NN | Bayes |
|---|---|---|---|
| Number of Topics | 100 | 50 | 50 |
| Number of Iterations | 1000 | 1000 | 1000 |
| alpha value | 0.90 | 0.95 | 0.95 |
| beta value | 0.10 | 0.10 | 0.01 |

Best parameters values for each classifier.

**LDA Parameter Values.** In order to apply the proposed LDA filter, a set of experiments are carried out to determine the best parameter values for each classifying algorithm, as shown in Table 3.

## Results

Table 4 shows the results achieved by each algorithm for each corpora, using both LDA and Bag of Words representation. The mean values correspond to the average Kappa measure obtained in each cross-validation process, while the "improv." column indicates whether there is a statistically proven improvement when the algorithm uses the LDA representation.

The SVM classifier shows a statistically better classification performance using LDA in only one of the tested corpus (Yahoo!). However, the difference between the mean values is very low in the case of Reuters and 20News corpora, being statistically irrelevant in the first corpus. This implies that the LDA representation offers in these cases a comparable performance.

**Table 4. Experiments results.**

| Data set | Classifier | BoW | | LDA | | Improv. |
|---|---|---|---|---|---|---|
| | | Mean | Sdev. | Mean | Sdev. | |
| Ohsumed C04 | SVM | 0.73 | $2.359 \cdot 10^{-2}$ | 0.67 | $2.569 \cdot 10^{-2}$ | NO |
| | k-NN | 0.15 | $4.945 \cdot 10^{-2}$ | 0.61 | $2.867 \cdot 10^{-2}$ | YES |
| | Bayes | 0.61 | $2.547 \cdot 10^{-2}$ | 0.28 | $1.89 \cdot 10^{-2}$ | NO |
| Reuters | SVM | 0.90 | $1.233 \cdot 10^{-2}$ | 0.91 | $1.214 \cdot 10^{-2}$ | – |
| | k-NN | 0.51 | $3.143 \cdot 10^{-2}$ | 0.89 | $1.370 \cdot 10^{-2}$ | YES |
| | Bayes | 0.68 | $1.928 \cdot 10^{-2}$ | 0.63 | $1.639 \cdot 10^{-2}$ | NO |
| 20News | SVM | 0.75 | $1.057 \cdot 10^{-2}$ | 0.72 | $9.15 \cdot 10^{-3}$ | NO |
| | k-NN | 0.42 | $1.139 \cdot 10^{-2}$ | 0.68 | $9.850 \cdot 10^{-3}$ | YES |
| | Bayes | 0.51 | $1.310 \cdot 10^{-2}$ | 0.49 | $1.503 \cdot 10^{-2}$ | NO |
| TREC Allele | SVM | 0.58 | $5.336 \cdot 10^{-2}$ | 0.47 | $5.631 \cdot 10^{-2}$ | NO |
| | k-NN | 0.00 | 0.000 | 0.430 | $5.755 \cdot 10^{-2}$ | YES |
| | Bayes | 0.46 | $3.000 \cdot 10^{-2}$ | 0.09 | $1.226 \cdot 10^{-2}$ | NO |
| Yelp Polarity | SVM | 0.67 | $2.687 \cdot 10^{-2}$ | 0.49 | $2.745 \cdot 10^{-2}$ | NO |
| | k-NN | 0.24 | $3.574 \cdot 10^{-2}$ | 0.42 | $3.350 \cdot 10^{-2}$ | YES |
| | Bayes | 0.57 | $3.690 \cdot 10^{-2}$ | 0.41 | $3.334 \cdot 10^{-2}$ | NO |
| Yahoo! | SVM | 0.43 | $1.549 \cdot 10^{-2}$ | 0.47 | $1.987 \cdot 10^{-2}$ | YES |
| | k-NN | 0.04 | $1.535 \cdot 10^{-2}$ | 0.38 | $2.222 \cdot 10^{-2}$ | YES |
| | Bayes | 0.43 | $1.816 \cdot 10^{-2}$ | 0.31 | $1.803 \cdot 10^{-2}$ | NO |

Results obtained.

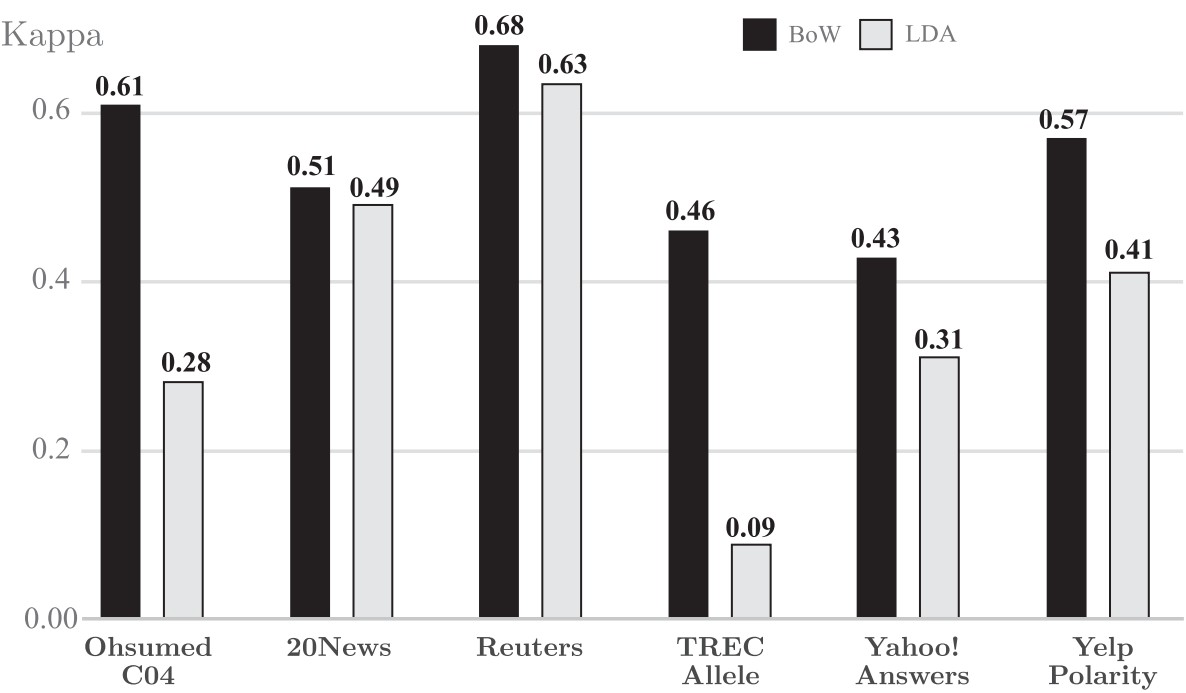

**Fig 7. Kappa results with Naive Bayes.** Ten fold cross validation mean Kappa results for Naive Bayes Classifier.

Naive Bayes is clearly negatively affected when using the LDA representation, as it can be seen in the classification results in all the tested corpus.

On the other hand, the k-NN classifier achieves a much better performance when using LDA across all the datasets. This is specially relevant in certain corpus like TREC, and Yahoo!, where the initial kappa values when using BoW are close to zero.

It should be noted that datasets OHSUMED C04 and TREC Allele are classified employing only two classes, indicating whether it is a relevant document or not. This negatively affects the Kappa values obtained by LDA, since this model uses a classification with a number of topics greater than two. As can be seen in Fig 7, a clear improvement is observed when LDA is applied to corpus that uses a larger number of classes such as 20News, Reuters and Yahoo! Answers stating that LDA offers better performance in a multi-class classification task.

In general, it can be stated that k-NN achieves good classification results using the LDA representation, but not for the vectors obtained by TF–IDF (Bag of Words). SVM and Naive Bayes, on the contrary, tend to get reliable results with a Bag of Words representation and ordinary for LDA. This is consistent with other related studies [10], where it is proven that the different classifiers are affected by LDA depending on the characteristics of the dataset.

In revised related studies [27], SVM and Naive Bayes also get more reliable results with BoW representations than LDA. In addition, the resultant dimmensionality also affects the performance of classifiers, with k-NN offering much better results when the feature matrix is not large, in contrast to SVM [28].

In addition, a study is carried out on the CPU usage time to train the classifiers measured in milliseconds. Fig 8 shows the SVM classifier spent time. A clear improvement can be seen in favor of LDA in all the corpus used, where LDA reduce to almost a third of the time used by BoW with the REUTERS corpus. In the other two cases, LDA completes the same task in less than one tenth of the time required using a BoW representation.

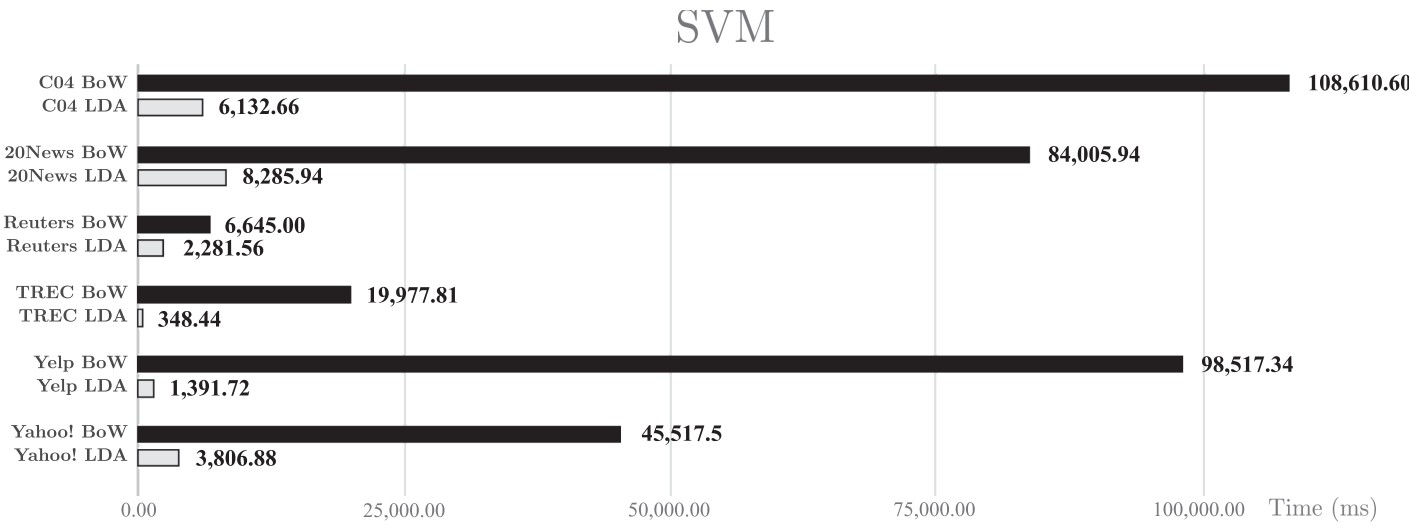

**Fig 8. CPU usage time with SVM.** CPU usage time in milliseconds for SVM classifier.

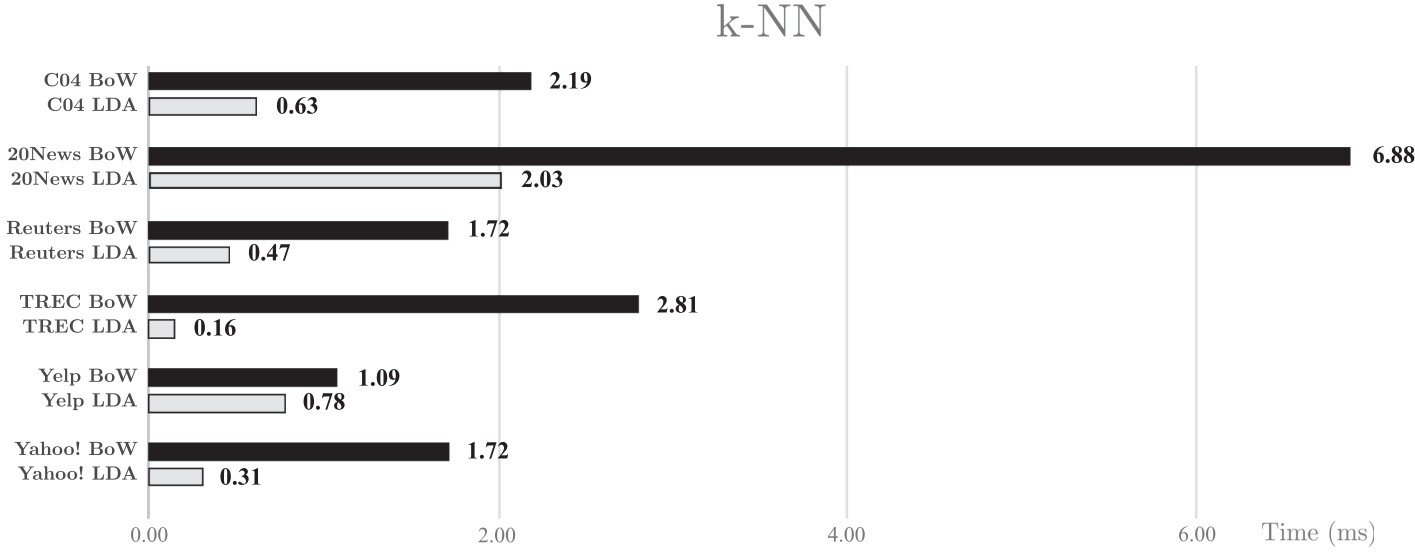

**Fig 9. CPU usage time with k-NN.** CPU usage time in milliseconds for k-NN classifier.

As can be seen in the results presented in Fig 9, the k-NN classifier benefits from the usage of LDA, completing the task three times faster than using BoW. This improvement in processing time supports the remarkably better performance values that are obtained using LDA with a k-NN classifier.

Finally, the processing times with Naive Bayes are presented in Fig 10. It can be seen that the execution times are clearly better when applying an LDA representation.

## Conclusions

In this work, we have proposed a text representation method based on LDA, implementing it as an extension to the Weka environment.

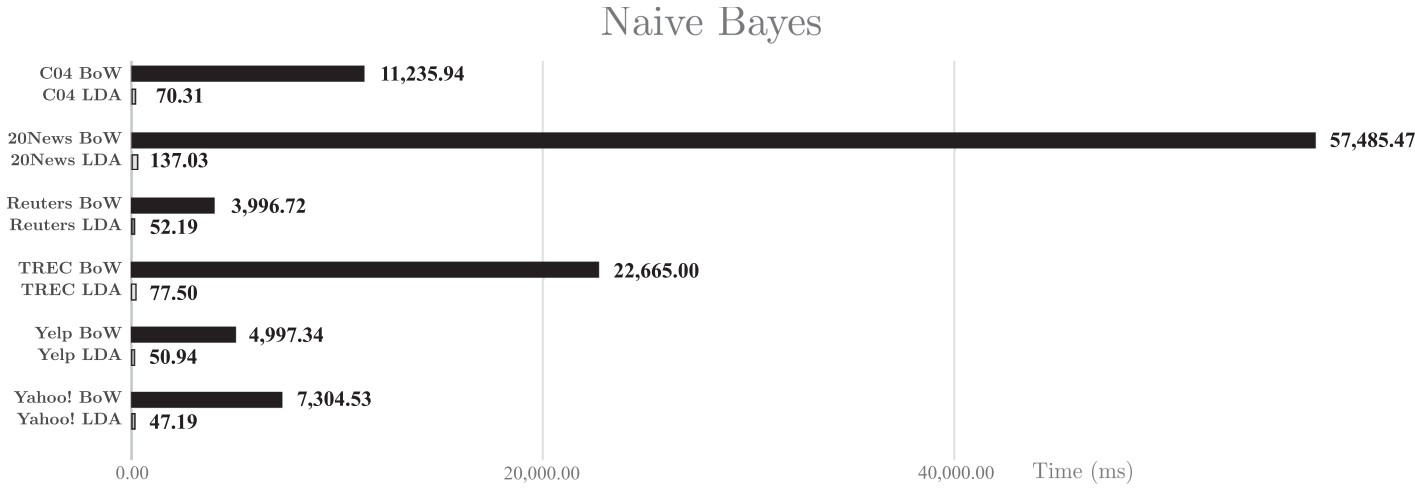

**Fig 10. CPU usage time with Naive Bayes.** CPU usage time in milliseconds for Naive Bayes classifier.

Taking into account the results obtained after the execution of different classifiers on different corpus using the implemented LDA representation, we conclude that LDA is a totally viable alternative to represent documents when using an SVM or k-NN classifier. LDA_filter offers similar or better values than those obtained with the commonly used text representation: the bag of words paradigm.

LDA representation offers a pronounced improvement in classification time over BoW. These results are supported by the fact that the LDA representation drastically reduces the dimensionality of the resultant feature space, improving the execution time of the classification algorithms.

In general, the performance of our proposed filter is very satisfactory. The inclusion of this filter in the Weka tool will help in future research to apply the LDA filter in other contexts or combining it with different techniques.

## Acknowledgments

SING group thanks CITI (Centro de Investigación, Transferencia e Innovación) from University of Vigo for hosting its IT infrastructure.

## Author Contributions

**Conceptualization:** P. Celard.

**Data curation:** P. Celard.

**Investigation:** P. Celard, A. Seara Vieira, E. L. Iglesias, L. Borrajo.

**Software:** P. Celard, A. Seara Vieira.

**Writing – original draft:** P. Celard.

**Writing – review & editing:** P. Celard, A. Seara Vieira.

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
