## [Decision Letter · Decision Letter 0]

13 Aug 2020

PONE-D-20-03130

LDA Filter: A Latent Dirichlet Allocation preprocess method for Weka

PLOS ONE

Dear Dr. Celard,

Thank you for submitting your manuscript to PLOS ONE. After careful consideration, we feel that it has merit but does not fully meet PLOS ONE’s publication criteria as it currently stands. Therefore, we invite you to submit a revised version of the manuscript that addresses the points raised during the review process.

The authors should particularly strive to address the following issues raised in review:

Can the authors propose an explanation for the fact that, for all datasets used, the LDA method systematically only improved the results of the kNN algorithm, but not SVM and NB?More data sets should be tested (possibly openly accessible ones as well), so the proposed LDA filter can be shown to provide significant improvements for a larger range of datasets.A broader literature survey should help to contextualize the problem statement, and should inform a discussion of the reasons why LDA improves text classification performance. In addition to source code, the authors should also share compile and install instructions to ensure reproducibility of the work described here, as well as documentation for the use of their plugin.

We look forward to receiving your revised manuscript.

Kind regards,

Marco Bonizzoni, Ph.D.

Academic Editor

PLOS ONE

Journal Requirements:

2.We note that the figures in your submission contain copyrighted images. All PLOS content is published under the Creative Commons Attribution License (CC BY 4.0), which means that the manuscript, images, and Supporting Information files will be freely available online, and any third party is permitted to access, download, copy, distribute, and use these materials in any way, even commercially, with proper attribution. For more information, see our copyright guidelines: http://journals.plos.org/plosone/s/licenses-and-copyright.

1.    You may seek permission from the original copyright holder of the figures to publish the content specifically under the CC BY 4.0 license.

3.Thank you for stating the following in the Funding Section of your manuscript:

[This work was partially supported by the Consellera de Educacion, Universidades e Formacion Profesional (Xunta de Galicia) under the scope of the strategic funding of ED431C2018/55-GRC Competitive Reference Group.]

 [The author(s) received no specific funding for this work.]

Reviewers' comments:

Reviewer's Responses to Questions

**Comments to the Author**

1. Is the manuscript technically sound, and do the data support the conclusions?

Reviewer #1: Partly

2. Has the statistical analysis been performed appropriately and rigorously? 

Reviewer #1: No

3. Have the authors made all data underlying the findings in their manuscript fully available?

Reviewer #1: Yes

4. Is the manuscript presented in an intelligible fashion and written in standard English?

Reviewer #1: Yes

5. Review Comments to the Author

Reviewer #1: * What are the main claims of the paper and how significant are they for the discipline?

The main objective is to create a filter for Weka, where text data could be transformed in the low dimension representation using LDA and show that the classification tasks using LDA representation are faster without compromising accuracy.

Using LDA for text representation has been in practice for several years now. So, there is no new research contribution. The only contribution from the authors is the creation of an LDA plugin for Weka.

* Are the claims properly placed in the context of the previous literature? Have the authors treated the literature fairly?

No, there are a lot of papers in the literature that uses LDA for information retrieval, search engines, text matching, text hashing, etc. The authors have just cited the base paper by Blei and the semi-supervised extension on LDA modeling. The literature survey is insufficient in the context of the problem statement.

* Do the data and analyses fully support the claims? If not, what other evidence is required?

The results of the experiments are presented in the paper. The authors have used LDA (by calling an API from the MALLET library) to build a filter for Weka. As per their own experimental results, the filter appears to be not useful for improved classification accuracy. In all the 3 datasets used in the experiments, the LDA method worked for just the kNN algorithm, but no reasoning provided. Also, there is no explanation provided for why the method didn't work for other algorithms (SVM and NB). Using just 3 datasets for the experiments seem insufficient to prove anything empirically. The authors claim "speed" improvement as a positive outcome, but it is not interesting as with any dimensional reduction technique speed improvement is obvious.

* PLOS ONE encourages authors to publish detailed protocols and algorithms as supporting information online. Do any particular methods used in the manuscript warrant such treatment? If a protocol is already provided, for example for a randomized controlled trial, are there any important deviations from it? If so, have the authors explained adequately why the deviations occurred?

Not applicable.

* If the paper is considered unsuitable for publication in its present form, does the study itself show sufficient potential that the authors should be encouraged to resubmit a revised version?

Yes, creation of an LDA filter for Weka is an useful contribution, but the authors should improve the LDA method to make the filter help in improving the classification accuracy. Some of the suggested amendments are:

+ More data sets to be tested.

+ LDA filter should be shown providing accuracy improvement for the majority of the datasets.

+ Thorough literature survey should be done to find cues for how to LDA for improving text classification performance.

+ The LDA tuning process can become costly if a grid search for parameters is done. So, a method for smart tuning should be suggested.

+ Source code is made available, but the preprocesed dataset and results are not available in public domain. Sufficient documentation of the source code should be provided with compile and install instructions.

* Are original data deposited in appropriate repositories and accession/version numbers provided for genes, proteins, mutants, diseases, etc.?

No. Data is not made available. The source code is made available in Github, but there are no instructions for compilation and testing. There is no documentation available for how to tune/use the plugin. The plugin is made available in Sourceforge, but no documentation either.

* Are details of the methodology sufficient to allow the experiments to be reproduced?

Yes, if we the use the plugin prebuilt for Weka (https://sourceforge.net/projects/weka-lda-filter/)

If we just follow the paper, it is not possible to reproduce the experiment.

* Is the manuscript well organized and written clearly enough to be accessible to non-specialists?

The paper is written like a technical report and not like a research article.

6. PLOS authors have the option to publish the peer review history of their article (what does this mean?). If published, this will include your full peer review and any attached files.

Reviewer #1: No

---

## [Author Response · Author response to Decision Letter 0]

3 Oct 2020

Editor: Thank you for giving us the opportunity to submit a revised draft of our manuscript. We appreciate the effort dedicated to provide feedback on our manuscript. We have incorporated all of your suggestions into our manuscript. 

Reviewer 1: Thank you for your insightful comments and valuable improvements. We have incorporated all of your suggestions into our manuscript.

---

## [Editor Report · Decision Letter 1]

20 Oct 2020

LDA Filter: A Latent Dirichlet Allocation preprocess method for Weka

PONE-D-20-03130R1

Dear Dr. Celard,

We are pleased to inform you that your revised manuscript satisfactorily addresses the issues raised during review and has been judged scientifically suitable for publication so it will be formally accepted for publication once it meets all outstanding technical requirements.

Kind regards,

Marco Bonizzoni, Ph.D.

Academic Editor

PLOS ONE
---

## [Editor Report · Acceptance letter]

26 Oct 2020

PONE-D-20-03130R1 

LDA Filter: A Latent Dirichlet Allocation preprocess method for Weka 

Dear Dr. Celard:

I'm pleased to inform you that your manuscript has been deemed suitable for publication in PLOS ONE. Congratulations! Your manuscript is now with our production department. 

Kind regards, 

on behalf of

Dr. Marco Bonizzoni 

Academic Editor

PLOS ONE